# Size-Dependent Switching in Thin Ferroelectric Films: Mathematical Aspects and Finite Element Simulation

Elena Veselova [1,*], Anna Maslovskaya [1] and Alexander Chebotarev [2]

1   Department of Mathematics and Computer Science, Amur State University, Ignatyevskoye Shosse, 21, Amur Region, 675027 Blagoveshchensk, Russia
2   Institute for Applied Mathematics, Far Eastern Branch of the Russian Academy of Sciences, Radio St., 7, 690041 Vladivostok, Russia
*   Correspondence: veselova.em@amursu.ru

**Abstract:** The paper is devoted to the theoretical analysis and numerical implementation of a mathematical model of a nonlinear reaction–diffusion system on the COMSOL Multiphysics platform. The applied problem of the computer simulation of polarization switching in thin ferroelectric films is considered. The model is based on the Landau–Ginzburg–Devonshire–Khalatnikov thermodynamic approach and formalized as an initial-boundary value problem for a semilinear parabolic partial differential equation. The theoretical foundations of the model were explained. The user interface design application was developed with COMSOL Multiphysics. A series of computational experiments was performed to study the ferroelectric hysteresis and temperature dependences of polarization on the example of a ferroelectric barium titanate film.

**Keywords:** reaction–diffusion system; Landau–Ginzburg–Devonshire–Khalatnikov model; finite elements method; polarization; ferroelectric hysteresis

## 1. Introduction

Currently, in the theory and practice of mathematical modeling and computer simulation, there is considerable interest in the study of reaction–diffusion processes. Equations of the "reaction-diffusion" type are used to describe space–time structures in nonlinear media of various natures. Examples of such applications can be cited in a wide variety of subject areas: chemistry, biology, economics, ecology, physics, the theory of heat and mass transfer, hydrodynamics, etc. [1–7].

So, for example, equations of the "reaction-diffusion" type have been used in problems of environmental forecasting when modeling the spread of pollutants [3], describing the processes of hemodynamics in large blood vessels in medicine [4], and modeling the behavior of prices in European and Asian options (the Black–Scholes model) in economics [5]. The Barkley model and systems with the Belousov–Zhabotinsky reaction determine the excitability and propagation of oscillations in biological media, Nagumo's equation formalizes the propagation of nerve impulses [1,6], and the reaction–diffusion model is used for description of oxygen transport in the brain [7].

The Fisher–Kolmogorov (or Kolmogorov–Petrovsky–Piskunov) equation has been used to describe chemical reactions, genetic systems, and population dynamics. Reaction–diffusion equations provide a fundamental basis for modeling many phenomena in physics and biology [1,8–12]: for quantum mechanical calculations of the probability density functions of particles (in the formulation of the Fokker–Planck, Kolmogorov, and Schrödinger equations) [1,8], for modeling heat and mass transfer and transport mechanisms of charged particles [2,9,11], for complex bacterial behavior [13], for simulation of electron beam induced charging processes in polar dielectrics [12,14], etc. The deterministic approach leads

to a fundamental description of the models of reaction–diffusion processes in the formulation of boundary and initial-boundary value problems for partial differential equations of parabolic type.

The Landau–Ginzburg model can also be referred to as the reaction–diffusion model, which is used to describe a large class of nonlinear phenomena in multicomponent and complex systems. For example, based on the Landau–Ginzburg equation, various aspects of signal propagation in cardiac tissue, superconductivity, superfluidity, and nonlinear optical systems have been described [15]. The model has also been applied to describe the processes in the physics of lasers, liquid crystals, and hydrodynamics, in quantum field theory [16].

In a series of works [17–26], the application of the Landau–Ginzburg model for studying the properties of ferroelectrics, modeling ferroelectric polarization switching, and hysteresis loops, has been considered in the framework of the Landau–Ginzburg–Devonshire ferroelectric theory. Mathematically, the space–time distribution of polarization can be described based on the Landau–Ginzburg–Devonshire–Khalatnikov thermodynamic model [17–19] in the formulation of the initial-boundary value problem for a partial differential equation of the "reaction-diffusion" type [21,22]. However, in practice, the problem can be simplified by reducing it to the initial value problem for an ordinary differential equation [25] or a boundary value problem for the ODE of second order [26]. In the first case, the spatial distribution of polarization in the volume of the sample and surface effects are not taken into account, which is crucial for low-dimensional structures. On the other hand, the use of a second order ODE provides the dependence of polarization on the coordinate, but assumes a stationary regime. The time dependence of the polarization-field is introduced artificially, assuming that the polarization has time to relax to the equilibrium state, which in the general case can be a rough approximation.

In the authors' works [22–24], the numerical aspects of the implementation of the 1D model of polarization switching have been considered; computer simulation of ferroelectric hysteresis has been performed based on the approaches of classical thermodynamic theory. A theoretical analysis of the model has been performed in [24], in which the existence and uniqueness of a weak solution to the initial-boundary value problem for the Landau–Khalatnikov equation has been established.

In the general case, the construction of analytical solutions for solving reaction–diffusion physical problems is associated with some serious difficulties, and practically, numerical methods and computing techniques are widely used. Recently, the finite element method has become increasingly popular among numerical methods for solving problems of mathematical physics. There is a wide range of software for the analysis of processes and phenomena based on the finite element approach. The functionality of specialized software products allows one to solve applied problems formalized by equations of the reaction–diffusion type. One of the effective platforms for the numerical implementation of this class of problems is represented by the COMSOL Multiphysics software. COMSOL Multiphysics is also a finite element-oriented modeling system. Using COMSOL tools, one can perform computer simulations of difficult-to-formalize systems and predict the characteristics of the analyzed processes.

The ultimate goal of the present study is to perform computer-assisted mathematical modeling of the polarization switching process as a nonlinear reaction–diffusion physical system using the tools of the COMSOL Multiphysics platform. Computational experiments, in particular, the study of the temperature dependence of hysteresis loops, are conducted dependent, for example, on studies of thin films of barium titanate. The important contributions of this work to the field are presented by the development of the mathematical basis and computational techniques for the implementation of the time-dependent cubic-quintic Landau–Khalatnikov model.

The rest of the paper is organized as follows. In Section 2 we introduce the governing equation of the reaction–diffusion Landau–Khalatnikov model for first-order phase transition ferroelectrics. Section 3 is devoted to the mathematical basis for the initial-boundary

value problem for the generalized Landau–Khalatnikov model, namely, the existence and uniqueness of the solution of the considered problem. Section 4 looks at the key steps of computer implementation of the model using COMSOL Multiphysics techniques. The results of numerical experiments and discussion are presented in Section 5. The user interface for scientific computing designed by COMSOL is described in Section 6. Section 7 contains concluding remarks.

## 2. Mathematical Model

Let us introduce into consideration the mathematical problem statement of the ferroelectric polarization model based on the Landau–Ginzburg–Devonshire thermodynamic theory. In the case of switching, polarization is realized along with one of its components and induced by an applied electric field, the mathematical model can be expressed as a two-dimensional (concerning spatial coordinates) initial-boundary value problem for the nonstationary Landau–Ginzburg–Devonshire–Khalatnikov equation:

$$\frac{\partial P}{\partial t} = D\Delta P + aP + bP^3 - cP^5 + \overline{E}(t), \ 0 < x < l, \ 0 < y < l, \ 0 < t \le \theta, \tag{1}$$

$$P|_{t=0} = P_0(x,y), \ 0 < x < l, \ 0 < y < l, \tag{2}$$

$$\frac{\partial P}{\partial \mathbf{n}}\bigg|_\Gamma = -\frac{P}{\lambda}, \ 0 < t \le \theta, \tag{3}$$

where $P(x,y,t)$ is the space–time distribution of the polarization in C/m$^2$; $\mathbf{n}$ is the outward normal vector to the boundary; $\Gamma$ is the boundary of the domain $(0,l) \times (0,l)$; $\lambda$ is the extrapolation length in m; $D$ is the thermodynamic parameter in m$^2$/s that makes sense of the combination of the gradient coefficient and the restoring force with sense and dimension of diffusion coefficient; $a = -A/\delta$, $b = -B/\delta$, and $c = C/\delta$ are the positive constants; $A$, $B$, and $C$ are the thermodynamic parameters depending on temperature $T$; $\delta$ is the kinetic coefficient in m $\times$s/F; $\theta$ is the observation time in s; $\overline{E}(t) = E(t)\nu/\delta$, and $E(t)$ is the applied electric field in V/m; and $\nu$ is the scaling factor.

To obtain dielectric hysteresis loops, the field is defined to be periodic. To be precise, we assume that polarization reversal is observed under the action of the sinusoidal electric field:

$$E(t) = E_{0x}\sin(\omega t) \tag{4}$$

where $E_{0x}$ is the amplitude of the electric field intensity in V/m; $\omega = 2\pi f$ is the radial frequency of the applied field in rad/s; and $f$ is the frequency of field oscillations in Hz.

Note that, Equation (1) can be classified as a semilinear (cubic-quintic) time-dependent equation of the reaction–diffusion type. Relation (1) is suitable for ferroelectrics with the first order of phase transition and it can be reduced to the cubic partial differential equation for ferroelectrics with the second order phase transition taking into account the sign of the coefficient $B < 0$.

## 3. Theoretical Justification of the Mathematical Model

Following [24], where a similar one-dimensional model has been considered, we transform problems (1)–(3) into the Cauchy problem for an equation with an operator coefficient.

Let us assume that $\Omega = (0,l) \times (0,l)$ is a bounded Lipschitz domain, $\Gamma = \partial\Omega$, $Q = \Omega \times (0,\theta)$. Here we denote the Lebesgue space by $L^p$, $1 \le p \le \infty$ (the space of $p$-integrable functions essentially bounded if $p = \infty$) and by $H^s$ the Sobolev space of $W_2^s$. We set $H = L^2(\Omega)$, $V = H^1(\Omega)$. By $V'$ we denote the dual space of $V$, $V \subset H = H' \subset V'$. By $L^s(0,\theta;X)$, $s \ge 1$, we denote the space of $p$-integrable on $(0,\theta)$ functions assuming values in a Banach space $X$. Accordingly, by $C([0,\theta];X)$ we denote to the space of continuous on functions assuming values in a Banach space $X$.

Let us define the operator $F: V \to V'$ that

$$(FP, v) = D \int_{\Omega} \nabla P \nabla v \, dx \, dy + \frac{D}{\lambda} \int_{\Gamma} P v \, d\Gamma, \; \forall P, v \in V$$

Further we multiply Equation (1) by the test function $v \in V$, then integrate over $\Omega$, using Green's formula for the first term on the right side and taking into account the boundary condition (3). As a result, we obtain a weak formulation of the initial-boundary value problems (1)–(3) in the form of the following Cauchy problem for an ordinary differential equation with an operator coefficient.

**Cauchy problem.** *Find a function $P \in L^s(0, \theta; V)$ such that*

$$P' + FP = aP + bP^3 - cP^5 + \overline{E}(t) \; a.e. \, on \, (0, \theta), \; P|_{t=0} = P_0 \tag{5}$$

It is easy to establish that if a weak solution (namely, solution to problem (5)) is smooth enough, then it will be a classical solution to the initial boundary-value problems (1)–(3).

The proof of the solvability of problem (5) can be carried out by the Galerkin method similarly to the case of the initial-boundary value problem with the homogeneous Dirichlet conditions ([27], Th. 1.1), [24]. The presence of the quantic term in the right-hand side of (1) allows us to obtain the necessary a priori estimates of the Galerkin approximations and prove the following result.

**Theorem 1.** *For $P_0 \in H$, $E \in L^2(Q)$, there exists a unique solution $P$ of (5) such that*

$$P \in C([0, \theta]; V) \cap L^6(Q), \; P' \in L^2(0, \theta; V') + L^{6/5}(Q).$$

## 4. Computational Details of the Model Implementation

The computational process corresponding to the model implementation can be visualized using the functional diagram shown in Figure 1.

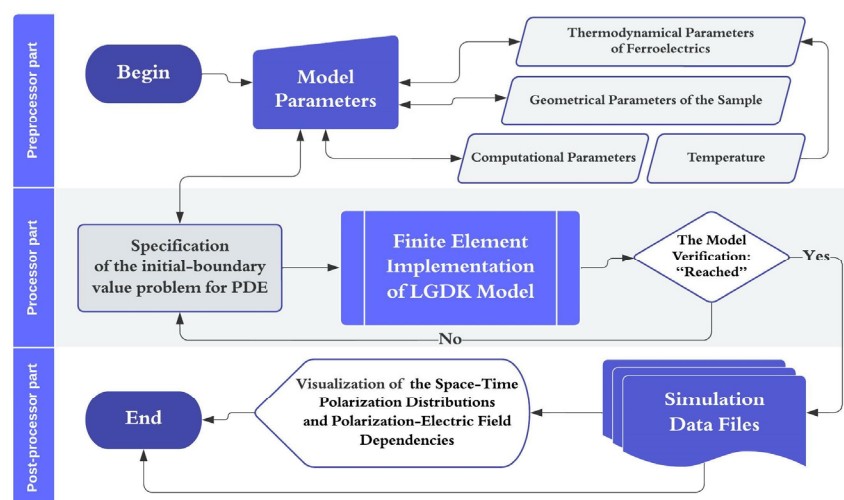

**Figure 1.** Functional block diagram of the model implementation.

To solve the initial-boundary value problem defined by (1)–(4), we use the COMSOL Multiphysics v5.0 platform. COMSOL is a software environment for simulation of a wide range of applied problems, which can be formalized by differential equations or systems of differential equations. Practical advantages of COMSOL Multiphysics are mostly the well-structured interface as well as user-friendly tools for model implementations. The finite element method is the main numerical method used in this software [28,29].

In this package, we can simulate both a separate process and related processes, taking into account the relationships between the quantities included in different equations of the mathematical model. The COMSOL platform is quite effective in computer modeling due to a joint interface and the integration of all model building processes in a convenient and visual environment. Regardless of the statement and complexity of the problem, the same functions and settings are used to initialize the model parameters. Furthermore, the platform tools are used to set and calculate custom systems of equations when solving non-standard problems that do not have a ready-made physical interface.

To solve the problem, we used the universal mathematical PDE (partial differential equation) interface of the COMSOL Multiphysics-based platform, which is not associated with any particular physics, namely, the coefficient form PDE (the coefficient form of the mathematical notation of the boundary value problem for PDE). The algorithm for the computer implementation of models (1)–(4) in the PDE interface consists of the following steps.

1.　Setting the model parameters, which means defining the dimensions of the model and the regime of the study.
2.　Defining and building the model geometry.
3.　Setting the parameters of the model and variables, setting the initial and boundary conditions, and determining the source function according to the mathematical formulation of the problem.
4.　Generation of a finite element mesh and splitting the solution domain into simple elements.
5.　Setting the solver settings, solving, and modifying the problem.
6.　Visualization and post-processing of the obtained results.

## 5. Implementation of the Model on the COMSOL Multiphysics Platform

Further, let us perform the implementation of the physical and mathematical model given by (1)–(4) using COMSOL Multiphysics. A series of computational experiments was carried out to study polarization switching in thin ferroelectric films.

Thin films of a typical barium titanate ($BaTiO_3$) ferroelectric were chosen as model objects. Barium titanate is a biaxial ferroelectric; nevertheless, the introduced approach and the geometry of the model make it possible to estimate the main characteristics of the polarization switching in c-domains. A simplified scheme of polarization reversal processes in the c-domains of a barium titanate ferroelectric sample under external field applied along the corresponding polar axis is demonstrated in Figure 2.

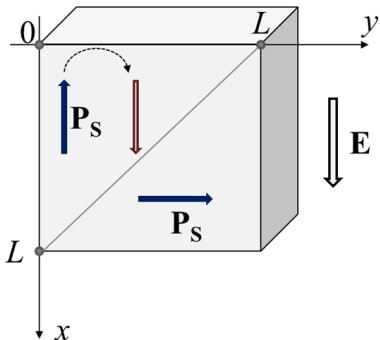

**Figure 2.** The diagram of polarization switching in c-domains of a $BaTiO_3$ ferroelectric sample.

To perform calculations, we initialize the constants and physical parameters of models (1) – (4), a complete list of which is presented in Table 1 (see [19,30] and references therein). The solution of the initial-boundary problem is presented by the space–time distribution of the polarization $P(x,y,t)$ in the computational domain.

**Table 1.** Model parameters.

| Parameter | Unit | Value |
|---|---|---|
| Sample linear size, $l$ | m | $50 \times 10^{-9}$ |
| Extrapolation length, $\lambda$ | m | $88 \times 10^{-9}$ |
| Diffusion coefficient, $D$ | m$^2$/s | $10^{-11}$ |
| $A/2$ | m/F | $3.34 \times (T - 381) \times 10^5$ |
| $B/4$ | m$^5$/(C$^2 \times$F) | $(3.6 \times (T - 448) - 202) \times 10^6$ |
| $C/6$ | m$^9$/(C$^4 \times$F) | $(-5.52 \times (T - 393) + 276) \times 10^7$ |
| Kinetic coefficient, $\delta$ | m$\times$s/F | $0.5 \times 10^4$ |
| Temperature, $T$ | K | 293 |
| Amplitude of the electric filed, $E_{0x}$ | V/m | $3 \times 10^6$–$6 \times 10^6$ |
| Frequency of field oscillations, $f$ | Hz | 100 |
| Observation time, $\theta$ | s | 0.015 |

Application of a sinusoidal field along one of the polarization components results in a periodic change in polarization. The polarization–electric field dependence $P(E)$ is defined as the time function of the average polarization over the sample area on the applied field. In this way, we can visualize hysteresis loops under variations of control model parameters such as simple thickness $l$, temperature $T$, frequency $f$, the amplitude of the electric field $E_{0x}$, etc. In general, the hysteresis loop under given thermodynamic conditions enables one to estimate important characteristics of polarization reversal processes such as the coercive field, remanent polarization, and spontaneous polarization.

The results of computations of the polarization–electric field dependence under variation of the simple thickness of the BaTiO$_3$ film $l$ are shown in Figure 3. These computational experiments indicate a significant dependence of the shape of the hysteresis loop on the linear size of the thin film.

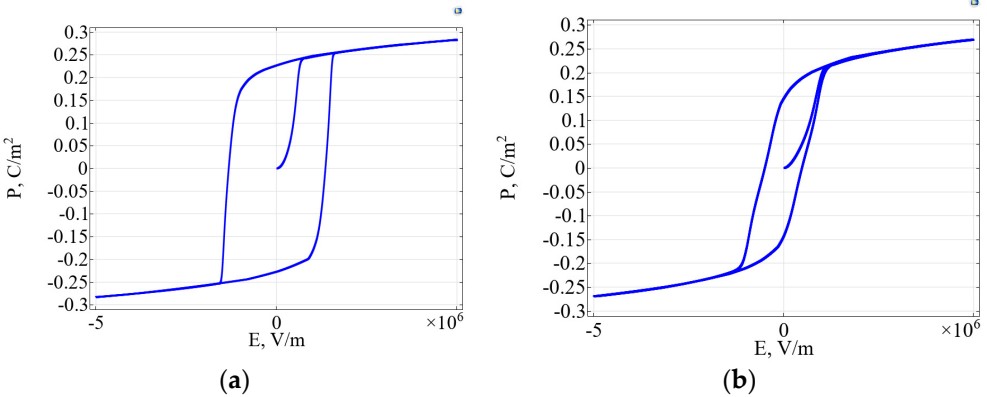

(**a**)  (**b**)

**Figure 3.** The polarization–electric field dependences under variation of thickness of the film: (**a**) $l = 50$ nm, (**b**) $l = 20$ nm.

If the thickness of BaTiO$_3$ films is smaller than 50 nm, the size effects of the hysteresis phenomenon can be observed. In this case, a decrease in the film thickness leads to deformation of the hysteresis loop with its subsequent disappearance.

Figure 4 illustrates the results of computer simulations of space–time distributions of polarization in BaTiO$_3$ films calculated at different values of sample thickness, in particular at $l = 0.1$ μm and $l = 10$ nm.

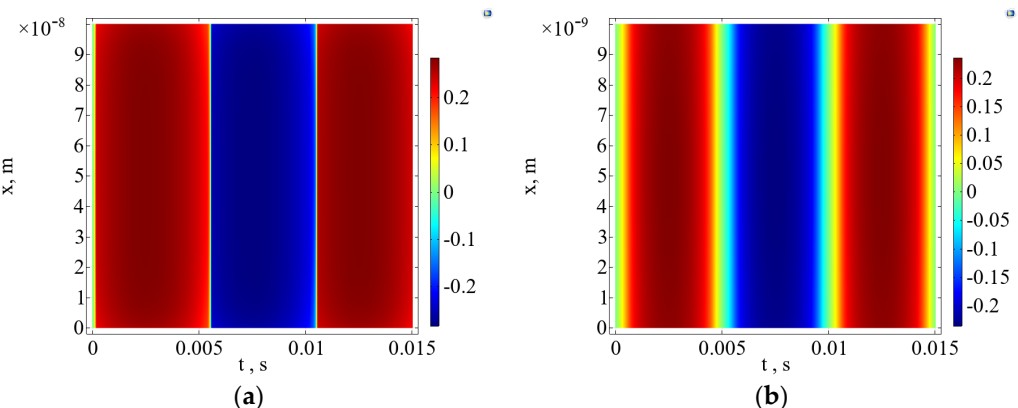

**Figure 4.** The space–time distributions of polarization in BaTiO$_3$ films computed at (**a**) $l$ = 0.1 μm and (**b**) $l$ = 10 nm.

These graphs show the temporal dynamics of polarization distribution through the sample thickness calculated at the fixed value of temperature $T$ = 293 K. It can be noted that the profile of temporal dependence of polarization is smeared out as the film thickness decreases. At a thickness of more than 100 nm, the shape is close to rectangular, but at thicknesses close to critical, it turns out to be significantly deformed and smoothed.

Here we can claim that the critical value of the film thickness is going to be 3–4 nm, below which the loop disappears. These findings obtained by computer simulations are consistent with the data obtained by the independent authors in [26] for the barium titanate nanowire. To estimate the hysteresis loops, a stationary analogue of the Landau–Ginzburg model in the formulation of the boundary value problem for an ordinary differential equation has been applied in [26].

The next interesting study, which can be performed with COMSOL Multiphysics, should be exploring polarization–electric field hysteresis dependence under variation of temperature. By inductive assumption, let us suppose that temperature is varied in the ferroelectric phase range. As a consequence, changes in temperature result in changes in the values of the corresponding thermodynamic parameters (see Table 1).

The results of computational experiments for BaTiO$_3$ films with a sample thickness $l$ = 50 nm are shown in Figure 5.

It can be noted that an increase in temperature leads to deformation and narrowing of the hysteresis loop, followed by its disappearance. The obtained results are consistent with studies of the temperature by polarization by the authors of the work [31].

Since both the temperature and the thickness of the ferroelectric sample are important to control parameters of the model, let us present the results of calculations of the temperature dependences of the remanent polarization $P_r$ (polarization values at zero applied field) and the coercive field $E_c$ (the electric field at which the macroscopic polarization disappears) in BaTiO$_3$ for different values of the sample thickness $l$. These dependences are approximately estimated with the use of calculated hysteresis loops and listed in Table 2. It should be pointed out that within the framework of this study, we do not aim to directly compare the simulation data with the experimental data. Nevertheless, the qualitative control characteristics of the model correspond to the experimental data. The experimental values of remanent polarization for a barium titanate single crystal approximately vary over the range $P_r$ = 0.23 ÷ 0.3 C/m$^2$ and the coercive field is evaluated as $E_c$ = 1 ÷ 2 10$^5$ V/m at room temperature [19].

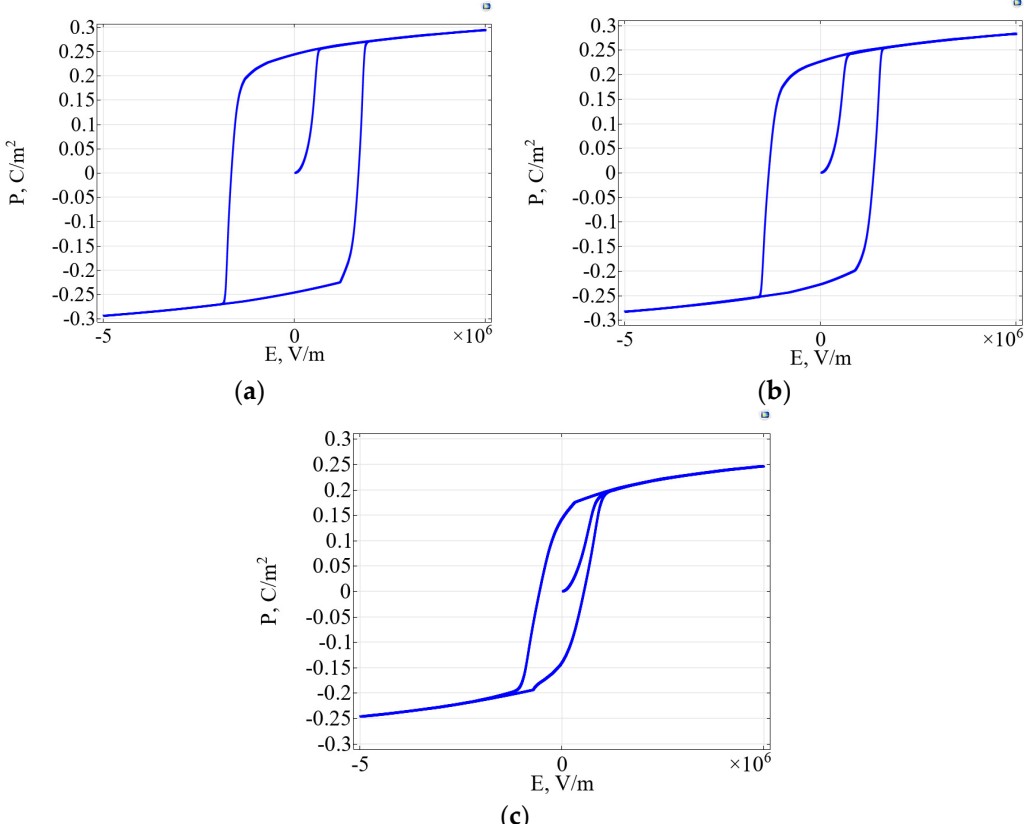

**Figure 5.** The polarization–electric field dependencies calculated at different values of the temperature: (**a**) $T = 278$ K, (**b**) $T = 293$ K, (**c**) $T = 353$ K.

**Table 2.** The remanent polarization $P_r$ and the coercive field $E_c$ under variation of sample thickness.

| Sample Linear Size, $l$ (nm) | Temperature, $T$ (K) | | | |
| | 293 | | 353 | |
| | $P_r$, C/m$^2$ | $E_c$, V/m | $P_r$, C/m$^2$ | $E_c$, V/m |
|---|---|---|---|---|
| 1 | 0.0002 | $2.50 \times 10^4$ | $16 \times 10^{-5}$ | $1.85 \times 10^4$ |
| 5 | 0.008 | $1.35 \times 10^5$ | 0.005 | $1.10 \times 10^5$ |
| 10 | 0.14 | $5.57 \times 10^5$ | 0.03 | $2.55 \times 10^5$ |
| 20 | 0.2 | $1.25 \times 10^6$ | 0.1 | $5.51 \times 10^5$ |
| 50 | 0.22 | $1.65 \times 10^6$ | 0.15 | $0.85 \times 10^6$ |
| 100 | 0.23 | $1.90 \times 10^6$ | 0.155 | $0.98 \times 10^6$ |
| 300 | 0.235 | $1.93 \times 10^6$ | 0.158 | $1.05 \times 10^6$ |

The results of calculations indicate that the value of the remanent polarization decreases with increasing temperature. In this case, the hysteresis loops become narrower and smaller. The loop deformation rate increases as the linear size of the film reaches a critical value.

Strictly speaking, the considered model can be classified as a quasi-two-dimensional model, since for multiaxial crystals it is necessary to take into account the expansion of the thermodynamic potential into the corresponding polarization components. Therefore, further development of the study will be focused on solving this applied problem in view of the specific configuration of the domain structure and the component representation of the polarization vector for a multiaxial crystal.

## 6. Developing User Interface Design Application for Mathematical Model with COMSOL Multiphysics

In order to make using COMSOL-oriented simulation more visual and convenient, a user interface application was developed with the tools provided by COMSOL Multiphysics. The application is based on a described mathematical model built on COMSOL and implemented by the finite element method. The user interface application is designed so that the user can analyze any model without spending a lot of time learning the program. COMSOL applications are specialized modeling tools that contain all the functionality of a model built in Model Builder mode but hide unnecessary information from the user.

The Application Builder is included with the core COMSOL Multiphysics platform and is accessible from the COMSOL Desktop GUI in Application Builder mode [32]. The Application Builder environment allows researchers to create ready-to-use intuitive user interfaces for their numerical models. The user of such an application operates only with the input data and meaningful results of the simulation, which does not require him to have a priori knowledge of the underlying model. The application development environment allows you to extend applications with user interfaces based on your models. Such a user interface may be a simplified version of the model, or it may contain parts of the input and output fields that need to be made available to the user.

The application with the user interface shown in Figure 6 was created using a form editor that does not require programming.

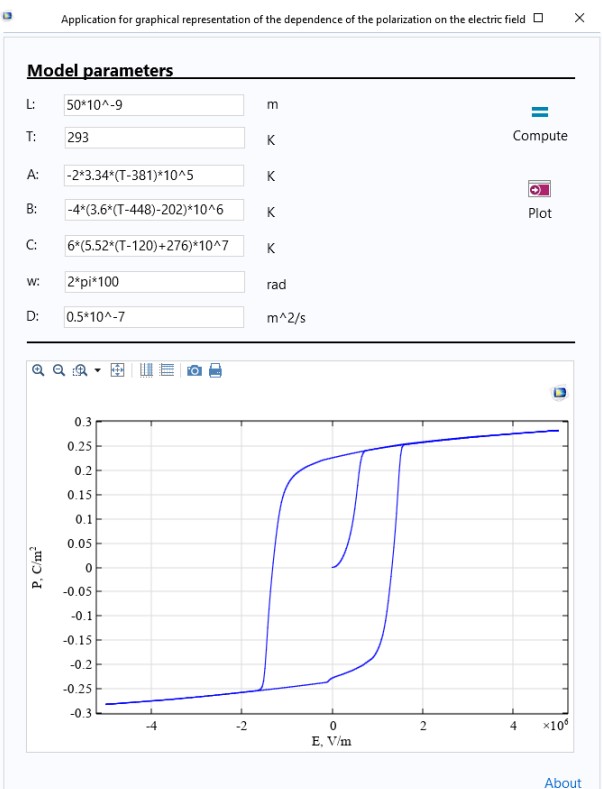

**Figure 6.** The general view of the user interface application for graphical representation of polarization–electric field dependencies.

The main control parameters of the model are placed on the main form of the application: sample thickness, temperature, thermodynamic constants, and diffusion coefficient. In the application run mode, the user can change the values of the parameters that define the model, and see the output of graphical results of the dependence of the polarization on the electric field by varying the values of the parameters. There are two function buttons on the main working screen of the application: Compute and Plot. Compute starts the process

of calculating the base model after changing the initial data. The Plot command allows us to display a graphical result after entering new values for the model parameters.

## 7. Conclusions

The complexity of the mathematical problem statements for reaction–diffusion models necessitates the use of finite element analysis software for their computer implementation. Specifically, the COMSOL Multiphysics platform provides effective tools for implementing mathematical models of nonlinear and semilinear reaction–diffusion processes which can be formalized by initial-boundary or/and boundary value problems. The finite-element simulation with COMSOL allows a researcher to avoid routine procedures for programming numerical algorithms.

In the present study, we consider the implementation of the Landau–Ginzburg–Devonshire–Khalatnikov thermodynamic model as an applied problem in the COMSOL environment. The considered model describes the process of polarization switching in ferroelectrics under an applied electric field. The mathematical model was formalized as a semilinear initial-boundary value problem and solved using the finite element method. The brief theoretical foundations for the analysis of the model were described. The obtained finding provides the theoretical basis for the numerical implementation of the thermodynamics model.

In addition, the user interface design application was developed in COMSOL Multiphysics in order to conduct computer simulation of the hysteresis loops of ferroelectrics under variation of control modeling parameters. Based on the computer implementation of the model, a series of computational experiments were carried out on thin films of a typical ferroelectric material made of barium titanate. The calculations allowed us to visualize the size effects of the polarization characteristics observed by independent authors. The computations of thickness- and temperature-dependent ferroelectric hysteresis loops of $BaTiO_3$ were examined. Our results indicate that a decrease in the film thickness leads to deformation of the hysteresis loop with its subsequent disappearance. The critical value of the film thickness is estimated to be 3–4 nm, below which the loop disappears.

The results of computer simulations of polarization–electric field dependencies under variation of temperature suggest that the maximum polarization value decreases with an increase in temperature followed by the complete disappearance of the hysteresis loop. Finally, additional studies are necessary to solve the applied problem in view of the specific configuration of the domain structure and the component representation of the polarization in the case of multiaxial crystals.

**Author Contributions:** Conceptualization, A.M.; methodology, E.V. and A.M.; software, E.V.; validation, E.V.; formal analysis, A.C.; investigation, A.C.; data curation, A.M.; writing—original draft preparation, E.V.; writing—review and editing, E.V. and A.M.; visualization, E.V.; supervision, A.M.; project administration, A.M.; funding acquisition, E.V., A.M. and A.C. All authors have read and agreed to the published version of the manuscript.

**Funding:** This work was supported by the Ministry of Science and Higher Education of the Russian Federation (project No. 122082400001-8).

**Institutional Review Board Statement:** Not applicable.

**Informed Consent Statement:** Not applicable.

**Data Availability Statement:** Not applicable.

**Conflicts of Interest:** The authors declare no conflict of interest.

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
