# Peer review of "Size-Dependent Switching in Thin Ferroelectric Films: Mathematical Aspects and Finite Element Simulation"

_computation, doi:10.3390/computation11010014_

Round 1
Reviewer 1 Report
I congratulate to authors for their clever work. I only suggest them that thy gave the numerical results tabularly.
Best Regards
Author Response
Response to Reviewer 1. Comment
The authors are grateful to the Reviewer for taking time to do the review and making the comment. The paper has been revised and the table has been added in order to remove the mentioned remark.
A PDF file with changes and corrections marked in red is attached. The new submission also contains changes related to the comments and remarks of other reviewers and Editor.
Please find below our replies and comments.
Sincerely yours,
Elena Veselova, Anna Maslovskaya and Alexander Chebotarev.
Point: I congratulate to authors for their clever work. I only suggest them that they gave the numerical results tabularly.
Response: The revised manuscript (i.e., Section 5) has been supplemented by a corresponding table instead of figure 6. In addition, this table includes the computational data of the coercive field Ec under variation of ferroelectric film thickness.

Reviewer 2 Report
There are aspects that are worthy of publication, however, this paper requires a major attention and careful considerations. Adequate revisions to the following points should be undertaken in order to justify recommendation for publication.
1. The advantages of the proposed method of this paper should be more highlighted. What is unique and novel? What is being done better?
2. Authors should clearly state the limitations of the proposed method in real applications.
3. New ideas with unique features compared to existing/published papers should be more highlighted. Kindly enrich the literature review of this paper by citing additional references related to the topic addressed, particularly on influence of disturbances, modeling errors, various uncertainties in the real systems. A relevant recent review are: Optimal experiment design for identification of ARX models with constrained output in non-Gaussian noise, Applied Mathematical Modelling; Surface effects on domain switching of a ferroelectric thin film under local mechanical load: A phase-field investigation, Journal of Applied Physics; Few shot cross equipment fault diagnosis method based on parameter optimization and feature metric, Measurement Science and Technology; Phenomenological calculation of the domain-size-dependent ferroelectric domain-wall velocity, Journal of the Korean Physical Society. It is necessary to make relation with the papers on this topic in Introduction section, and in that way, point out other contemporary approaches and possibilities. I believe this would further strengthen the introduction and lend support to the methodology applied in general.
4. Authors should argue their choice of the performance evaluation indicators.
5. Have the authors experimented with other sets of values? What are the sensitivities of these parameters on the results?
6. Explain the feasibility of the results from the implementation and computational point of view.
Author Response
Response to Reviewer 2 Comments
The authors are grateful to the Reviewer for taking time to review and making the comments. The paper has been revised and some issues have been clarified in more detail in order to remove the mentioned remarks. A PDF file with changes and corrections marked in red is attached. The new submission also contains changes related to the comments and remarks of other reviewers and Editor.
Nevertheless, part of the review includes questions that are answered in the article, and the information required for submission seems to the authors to be redundant for presenting the subject of consideration within the framework of one article. Please find below our replies and comments.
Sincerely yours,
Elena Veselova, Anna Maslovskaya and Alexander Chebotarev.

Reviewer 3 Report
The paper reports the results of the mathematical modelling of polarization switching in ferroelectrics based on the Landau-Ginzburg-Devonshire-Khalatnikov thermodynamic approach from both theoretical and computational points of view. The time-dependent cubic-quintic Ginzburg-Landau model is paid close attention by many researchers. This model is used to describe the key characteristics of ferroelectrics which underline increasingly important and promising applications of these materials. Unfortunately, nowadays, reported studies do not cover theoretical and numerical analysis of the generalized Landau-Khalatnikov model in a complete sense, especially for ferroelectrics with the first-order phase transition. In these terms, the topic is relevant.
The paper presents some interesting and novel findings that will be an up-to-date contribution to the field of mathematical modeling and numerical simulation of nonlinear physical systems. The theoretical foundations of the Landau-Khalatnikov model were explained. The numerical solution was obtained using the finite element method and the user interface design application was developed by COMSOL Multiphysics. A series of computational experiments was performed to study the ferroelectric hysteresis and temperature polarization dependences for a ferroelectric barium titanate film. The paper is well organized and the study seems to be very promising for future investigations.
The manuscript can be accepted for publication provided the authors go through some minor revisions in the view of the following comments. Most comments are suggested to improve the reader’s comprehension of the text.
1. It is not indicated in which case the solution of the Cauchy problem (4) (p. 4, line 138) for equations with operator coefficient will be a classical solution of the initial boundary value problem (1) – (3). Some comments concerning this issue should be provided.
2. The order of numbering is inconsistent. In the text of the article there are two formulas with the number 4 (lines 113 and 138).
3. As far as COMSOL Multiphysics is a commercial package, the license number should be specified in the manuscript.
4. As the authors noted in the Introduction part, there are several approaches to modeling of polarization properties based on the Landau-Khalatnikov equation, including those based on ODEs. It should be commented in the paper on why the use of PDE is important, in particular for ferroelectric thin films.
Author Response
Response to Reviewer 3 Comments
The authors are grateful to the Reviewer for taking time to review and making the comments. The paper has been revised and some issues have been clarified in more detail in order to remove the mentioned remarks.
A PDF file with changes and corrections marked in red is attached. The new submission also contains changes related to the comments and remarks of other reviewers and Editor.
Please find below our replies and comments.
Sincerely yours,
Elena Veselova, Anna Maslovskaya and Alexander Chebotarev.
